# Ultrasound Doppler Findings in Fetal Vascular Malperfusion Due to Umbilical Cord Abnormalities: A Pilot Case Predictive for Cerebral Palsy

**DOI:** 10.3390/diagnostics13182977

**Published:** 2023-09-18

**Authors:** Shota Saji, Junichi Hasegawa, Junki Koike, Misato Takatsuki, Natsumi Furuya, Nao Suzuki

**Affiliations:** 1Department of Obstetrics and Gynecology, St. Marianna University School of Medicine, Kawasaki 216-8511, Japan; seaspoon32@gmail.com (S.S.); natsumifuruya@outlook.jp (N.F.); nao@marianna-u.ac.jp (N.S.); 2Department of Pathology, St. Marianna University School of Medicine, Kawasaki 216-8511, Japan; j2koike@marianna-u.ac.jp (J.K.); misato.takatsuki@marianna-u.ac.jp (M.T.)

**Keywords:** single umbilical artery, long umbilical cord, umbilical cord entanglement, fetal thrombo-vasculopathy, fetal vascular malperfusion

## Abstract

Fetal Vascular Malperfusion (FVM), a pathologic condition in the feto-placental circulation, is a chronic vaso-occlusive disorder in the umbilical venous blood flow. Microthrombi are caused by the umbilical cord’s blood flow deficiency in a congested umbilical vein, which also causes microvascular damage to fetal organs, especially the brain, resulting in cerebral palsy. Thrombo-occlusive events also adversely affect the upstream chorionic or stem vessels in the placenta, resulting in fetal growth restriction and fetal hypoxia. An umbilical cord structural anomaly or multiple entanglements may involve FVM. In the present report, we demonstrate a case of FVM caused by multiple umbilical cord abnormalities obtained from antenatal ultrasound Doppler findings, and we also discuss FVM, which is chronically associated with CP, comparing the ultrasonographic findings to the pathologic findings.

Fetal Vascular Malperfusion (FVM) is a chronic vaso-occlusive condition of the umbilical venous flow, involving the placental villous ramification secondary to thrombo-occlusive conditions in the upstream chorionic or stem vessels in the placenta [1,2,3]. Decreased Wharton’s jelly, a narrow or long umbilical cord, and a hyper-coiled cord may be associated with such conditions. In affected cases of FVM, the contiguous terminal villi lose villus capillaries, resulting in the formation of avascular villi [1]. Consequently, these pathological changes involve fetal growth restriction and/or fetal hypoxia.

On the other hand, a vaso-occlusive condition in FVM causes microthrombi in the congested umbilical vein. Secondly, these thrombi cause microvascular damage to fetal organs. A previous study demonstrated that umbilical cord abnormalities, which can obstruct the umbilical cord blood flow, were significantly increased in neonates with placental FVM, who later developed cerebral palsy (CP) and other forms of long-term neurologic impairment [3].

At present, FVM represents a pathophysiology that is evaluated retrospectively from the condition of the neonate and the delivered placental findings. There are no previous studies on ultrasound findings that considered the possibility of FVM during pregnancy. Our previous research not only used antenatal ultrasounds to detect umbilical cord abnormalities and assess the risk of fetal hypoxia at delivery, but also to capture pathological changes in the umbilical cord and placenta [4,5]. We believe that prenatal evaluation is necessary not only to prevent the neurological impairment of the fetus due to acute hypoxia involving umbilical cord abnormalities, but also to assess for chronic adverse in utero effects.

In the present report, we experienced a case of FVM obtained via antenatal ultrasound Doppler findings. The purpose of the report is to clarify the pathogenesis and pathophysiology of FVM, comparing the ultrasonographic findings to the pathologic findings.

A nulliparous pregnant woman was diagnosed with a single umbilical artery (SUA) at a routine ultrasound checkup at 20 weeks of gestation, with a follow-up scan to determine fetal growth. At 35 weeks of gestation, she was referred to our perinatal center due to decreased fetal movement. The estimated fetal weight was 1944 g (−1.6 SD) with normal amniotic fluid volume (maximum vertical pocket: 3.7 cm). The placenta was located on the fundus of the uterus without any abnormal findings on the gray scale ultrasound. The torsion and insertion of the umbilical cord was within normal values, and the SUA was observed. Doppler velocimetry indicated the following values: umbilical arterial (UA) resistance index (RI) of 0.56, umbilical venous (UV) velocity of 5.3 cm/s without pulsation, and umbilical venous flow volume at the free loop (UVFV) of 108.7 mL/min (Figure 1A,B). The fetal middle cerebral artery (MCA) peak systolic velocity (PSV) was 101 cm/s (1.95 MoM), with an RI of 0.68, and the fetal descending aorta PSV was 176 cm/s (Figure 1C). The ultrasound equipment used in this case was an Aplio i700 with a 1–8 MHz (i8Cx1 probe PVI-475BX) convex probe (Canon Medical Systems, Otawara, Japan).

Fetal movement and fetal breathing were undetectable. CTG tracing showed low variability in the fetal heart baseline without acceleration and with repeated variable decelerations. Additionally, fetal bradycardia occurred, and an emergency cesarean section was performed (Figure 2). A small for gestational age infant (1786 g, with Apgar scores of 6/8 at 1/5 min and umbilical arterial blood of pH 6.91) with five loops of nuchal cord was born. The cell blood count of the neonate was 12.6 g/dL of hemoglobin and 216/nL of platelets. After the treatment of nasal continuous positive airway pressure therapy in the NICU, neurological complications or brain hemorrhage did not occur.

The placenta weighed 499 g with a normal placental cord insertion. The umbilical cord was thin and excessively long at 127 cm. One place with strong congestion of the umbilical vein was investigated (Figure 3). Using the microscopic evaluation of the umbilical cord, the presence of a formed umbilical veinous thrombus was confirmed. The remnant of one umbilical artery and arterial thrombosis were not found (Figure 4A). In the placenta, vascular thrombi were identified microscopically within the vascular lumina of the chorionic blood vessels. Intramural fibrin deposition was observed under the chorionic plate at the umbilical cord insertion (Figure 4B). Avascular villi were scattered in places, not over a large area (Figure 4C). There was no diffuse pattern of villi with increased syncytial knots and intervillous fibrin.

In the present case, an excessively long umbilical cord and its entanglement with intrauterine growth restriction resulted in a non-reassuring fetal status, which required an emergency cesarean section. The SUA was also associated with these complications. So far, no neurological injury has occurred in this neonate. However, in our present case, the pathologic findings were suggestive of FVM due to umbilical cord abnormalities. Further, the ultrasound Doppler findings during that pregnancy would have been consistent with those findings.

Regarding the interpretation of the Doppler findings, first, the SUA was related to the reduction in the UA-RI. A normal umbilical cord has two umbilical arteries, which branch out at the internal iliac artery and anastomose around the umbilical cord placental insertion (Hyrtl anastomosis). This structure gradually equalizes the blood flow resistance in the two umbilical arteries even when there is blood flow insufficiency due to external force. According to the histologic findings, an agenesis type of SUA, or the disappearance of an umbilical artery due to an occlusive event in the earlier gestation, is supposable rather than a recent occlusion. Therefore, in order to normalize the blood perfusion from the fetus to the placenta, a low RI in the umbilical artery was physiologically plausible.

Second, in the present case, a markedly increased MCA-PSV was observed even without fetal anemia. It is well-known that the measurement of MCA-PSV has been reported to predict the presence of moderate or severe fetal anemia with 100% sensitivity, while having a false positive rate of 12% [6]. High MCA-PSV in fetal anemia is considered a physiological change secondary to a raised cardiac output and decreased blood viscosity (6). On the other hand, it is also known that in growth-restricted fetuses, the increased MCA-PSV reflects an increased blood flow to the brain through an elevated left cardiac output and an increased placental vascular resistance [7].

Since the MCA-PSV and fetal aortic PSV in our case increased, we assume that this was caused by an event that resulted in elevated left cardiac output. The obtained ultrasonographic findings suggest that high MCA-PSV is not only the ultrasound marker for fetal anemia but is also significant of fetal malperfusion in feto-placental circulation, especially alongside umbilical cord occlusive abnormalities, which suggest a decrease in preload due to the umbilical veinous blood flow insufficiency. In fact, the umbilical veinous flow velocity and flow volume were decreased. The absence of the umbilical venous pulsation also supports this condition.

We think that a decreased umbilical venous flow with elevated MCA-PSV of a fetus with umbilical cord abnormalities may be related to pathologies associated with FVM. In cases with umbilical cord abnormality, the measurement of such Doppler parameters might be used to detect the FVM condition and further fetal neurological complications. Further prospective observational cohort studies are needed.

Finally, FVM is a term established by the Amsterdam International Consensus group of placental pathologists in 2015 to characterize a group of lesions previously described under the headings of fetal vascular obstructive lesions, fetal thrombotic vasculopathy, fetal vascular thrombi, and extensive avascular villi [8]. These complications may be attributed to several mechanisms, including altered feto-placental circulation, generalized systemic activation of the fetal coagulation system, and direct embolism of the placental thrombi to the fetus [3].

In the present case, the presence of umbilical veinous thrombus and scattered avascular villi were the findings of a microscopic evaluation of the placenta, which was compatible with FVM. However, the placental pathology did not show a high-grade pattern. In FVM, often the syncytial knots, composed of packed syncytiotrophoblast nuclei displaying heavily condensed chromatin, reflect accelerated villous maturation, but in the present case, there was no pathological increase. Although non-reactive fetal heart tracing associated with umbilical cord abnormalities was observed, the prognosis of the neonate was not that bad. Considering the fetal growth and low-grade pathologic findings, the umbilical cord blood deficiency might have occurred sub-acutely.

It is reported that growth-restricted fetuses diagnosed with a birth weight below the third percentile exhibiting abnormal umbilical cord insertion and an SUA are at a high risk of poor neurological outcomes, including cerebral palsy and/or developmental disorders [9]. FVM can cause abnormal clinical features such as growth restriction and an abnormal fetal heart rate, concomitant with an abnormal pathological condition.

As limitations of the pathological diagnosis of FVM, it has been difficult to distinguish whether the findings of FVM are made by observing thrombophilia or an obstruction before stillbirth, or involutional or degenerative changes following fetal demise [10]. However, we believe that the ultrasound diagnosis of umbilical cord and placental abnormalities, as well as additional changes in the Doppler findings, will help to predict the pathophysiology of FVM antenatally.

In conclusion, clinical relations between umbilical cord abnormalities and FVM, which is chronically associated with CP, were demonstrated. Although FVM is only currently diagnosed postnatally, we believe that the ultrasound diagnosis of umbilical cord abnormality and, in particular, Doppler velocimetry will enable the determination of the FVM condition antenatally. As a first step, we demonstrated that a decreased umbilical venous flow with elevated MCA-PSV of a fetus with umbilical cord abnormalities may be related to pathologies associated with FVM. Further case–control studies are needed, but this report may help to predict neonatal outcomes for infants with chronic cord abnormalities.

## Figures and Tables

**Figure 1 diagnostics-13-02977-f001:**
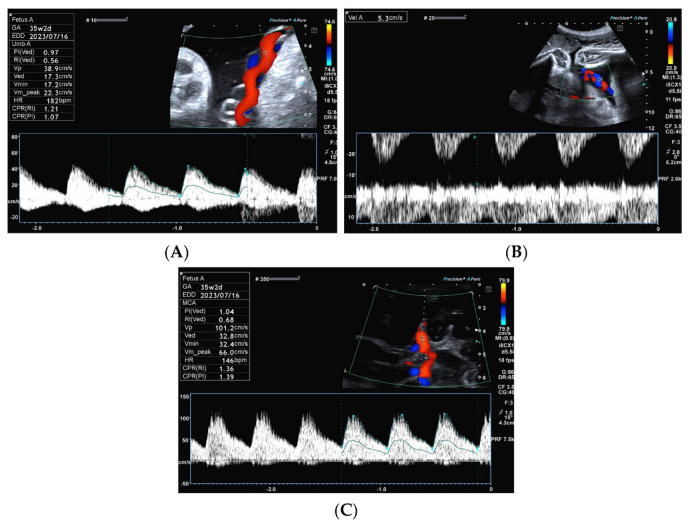
Transabdominal Doppler findings: (**A**) The systolic phase of the normal umbilical artery flow. (**B**) Umbilical venous flows; velocity was 5.3 cm/s without pulsation. The high peak systolic velocity of the middle cerebral artery was 101.2 cm/s or 1.95 MoM (**C**).

**Figure 2 diagnostics-13-02977-f002:**
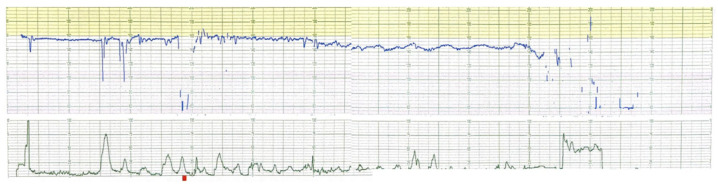
Cardiotocogram before cesarean section: low variability and fetal bradycardia were observed.

**Figure 3 diagnostics-13-02977-f003:**
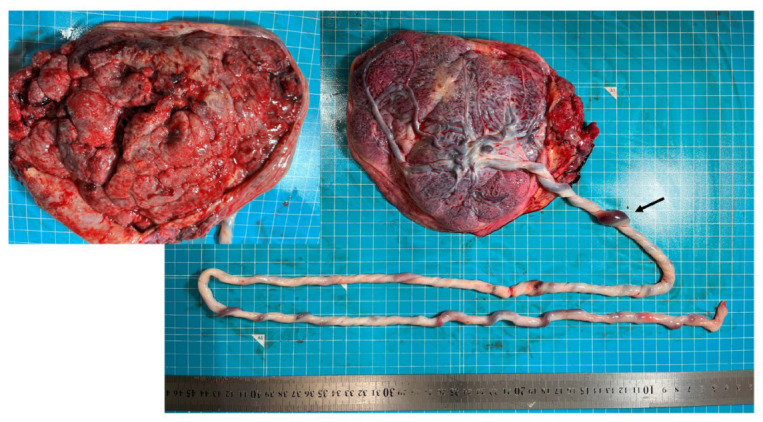
Macroscopic findings of the placenta: the umbilical cord was thin and excessively long at 127 cm. One place with strong congestion of the umbilical vein was suspected (arrow).

**Figure 4 diagnostics-13-02977-f004:**
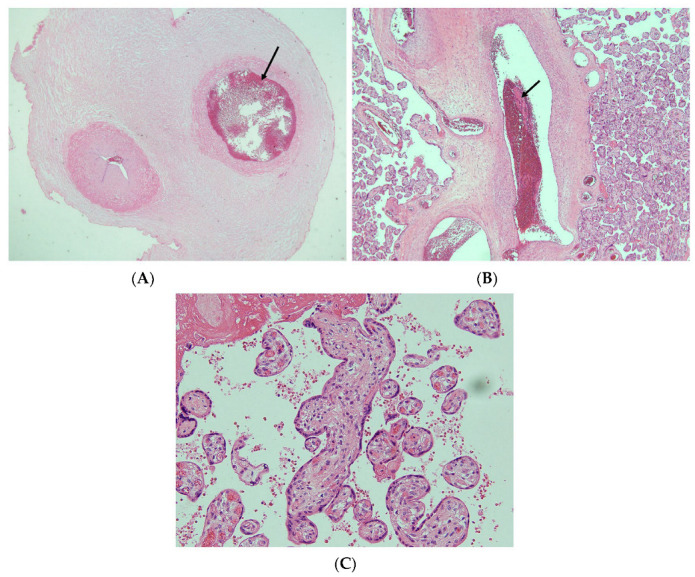
Histological findings of the placenta: (**A**) A single umbilical artery without remnants of one artery or thrombosis of the umbilical vein (arrow) were observed. (Hematoxylin–eosin, original magnification ×2). (**B**) Intramural fibrin deposition (arrow) was observed under the chorionic plate at the placental cord insertion (hematoxylin–eosin, original magnification ×10). (**C**) Avascular villi were scattered in places, not over a large area (hematoxylin–eosin, original magnification ×20).

## Data Availability

Data sharing not applicable—no new data generated.

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
