# Peer review of "Ultrasound Doppler Findings in Fetal Vascular Malperfusion Due to Umbilical Cord Abnormalities: A Pilot Case Predictive for Cerebral Palsy"

_diagnostics, 2023, doi:10.3390/diagnostics13182977_

Round 1
Reviewer 1 Report
I read with great interest the study conducted by Shota Saji et al., this article presents a case study on fetal vascular malperfusion (FVM) caused by umbilical cord abnormalities. The study examines the ultrasound Doppler findings and compares them to the pathological findings. The results suggest that Doppler measurements can help detect FVM in cases of umbilical cord abnormalities and predict fetal neurological complications.
In general, is well written.
I have a few comments:
The introduction only briefly mentions the case study and does not provide a comprehensive overview of fetal vascular malperfusion (FVM) or the significance of ultrasound Doppler findings in diagnosing FVM. Additionally, the introduction does not cite any references to support the information provided.
The information is presented in a clear and organized manner. However, it should be noted that the information provided is a compilation of excerpts from different sources, and the flow of the report could be improved.
The scientific soundness of the information provided cannot be determined as it is a compilation of excerpts from different sources and does not include a comprehensive analysis or discussion of the case study.
Although the pathological findings are clear, in order to demonstrate that the decrease in umbilical venous flow with elevated MCA-PSV is associated with FVM, cases and controls and the correlation with the pathological findings are necessary. The authors could include this and be clearer in the conclusion.
Please improve the quality of the figures.
Author Response
Answer to reviewer 1
I read with great interest the study conducted by Shota Saji et al., this article presents a case study on fetal vascular malperfusion (FVM) caused by umbilical cord abnormalities. The study examines the ultrasound Doppler findings and compares them to the pathological findings. The results suggest that Doppler measurements can help detect FVM in cases of umbilical cord abnormalities and predict fetal neurological complications.
In general, is well written.
A) Thank you for taking your time to review our paper and valuable comments. According to reviewers’ comments, we revised paper and answered to comments. We would appreciate if you could re-review kindly.
I have a few comments:
The introduction only briefly mentions the case study and does not provide a comprehensive overview of fetal vascular malperfusion (FVM) or the significance of ultrasound Doppler findings in diagnosing FVM. Additionally, the introduction does not cite any references to support the information provided.
The information is presented in a clear and organized manner. However, it should be noted that the information provided is a compilation of excerpts from different sources, and the flow of the report could be improved.
The scientific soundness of the information provided cannot be determined as it is a compilation of excerpts from different sources and does not include a comprehensive analysis or discussion of the case study.
A) Thank you for your nice advice. We added following sentences.
At present, FVM represents a pathophysiology that is evaluated retrospectively from the condition of the neonate and delivered placental findings. There are no previous studies on ultrasound findings that considered the possibility of FVM during pregnancy. Our previous research not only used antenatal ultrasound to detect umbilical cord abnormalities and assess the risk of fetal hypoxia at delivery, but also to capture pathological changes in the umbilical cord and placenta (4, 5). We believe that prenatal evaluation is necessary not only to prevent neurological impairment of the fetus due to acute hypoxia involved by umbilical cord abnormalities, but also to assess for chronic adverse in utero effects.
Although the pathological findings are clear, in order to demonstrate that the decrease in umbilical venous flow with elevated MCA-PSV is associated with FVM, cases and controls and the correlation with the pathological findings are necessary. The authors could include this and be clearer in the conclusion.
A) We understand your opinion. Now, we are engaging into comparison between the antenatal ultrasound and pathological findings to detect chronic condition associated with long term neurological complication. At the first, we experienced the case suspecting FVM during antenatal period. Referred to findings in this case, we are going to collect further data. Conclusion are revised as follows.
In conclusion, clinical relations between umbilical cord abnormalities and FVM which is chronically associable with CP were demonstrated. Nowadays, though FVM is only pathologic diagnosis postnatally, we believe ultrasound diagnosis of umbilical cord abnormality and particular Doppler velocimetry will suspect FVM condition antenatally. As a first step, we demonstrated decreased umbilical venous flow with elevated MCA-PSV of a fetus with umbilical cord abnormalities is possible of pathologies associated with FVM. Further case-control studies are needed, but this report may help predict neonatal outcomes for infants with chronic cord abnormalities.
Please improve the quality of the figures.
A) They are revised.
Reviewer 2 Report
Clinical case is well described in detail. It has long been known that the single umbilical artery is associated with fetal malformation. But it was not clear enough how it could be explained. This observation well reflects the hemodynamic disturbances that occur with this type of defect. Instrumental data are given. I think after making minor corrections and additions, the article may be published.
There are minor errors in the text. The text of the article should be corrected a bit.
Author Response
Answer to reviewer 2
A) Thank you for taking your time to review our paper and valuable comments. According to reviewers’ comments, we revised paper and answered to comments. We would appreciate if you could re-review kindly.
Clinical case is well described in detail. It has long been known that the single umbilical artery is associated with fetal malformation. But it was not clear enough how it could be explained. This observation well reflects the hemodynamic disturbances that occur with this type of defect. Instrumental data are given. I think after making minor corrections and additions, the article may be published.
A) Thank you very much. We revised throughout manuscript to brush up as you and reviewer’s suggestions. Language is also edited by native speaker before re-submission.